# Skeletal-Vascular Interactions in Bone Development, Homeostasis, and Pathological Destruction

**DOI:** 10.3390/ijms241310912

**Published:** 2023-06-30

**Authors:** Haruhisa Watanabe, Nako Maishi, Marie Hoshi-Numahata, Mai Nishiura, Atsuko Nakanishi-Kimura, Kyoko Hida, Tadahiro Iimura

**Affiliations:** 1Department of Pharmacology, Faculty and Graduate School of Dental Medicine, Hokkaido University, N13 W7, Sapporo 060-8586, Hokkaido, Japan; hwatanabe@den.hokudai.ac.jp (H.W.); hoshi-numahata.marie@den.hokudai.ac.jp (M.H.-N.); nishimai@den.hokudai.ac.jp (M.N.); anakanishi@den.hokudai.ac.jp (A.N.-K.); 2Department of Vascular Biology, Faculty and Graduate School of Dental Medicine, Hokkaido University, N13 W7, Sapporo 060-8586, Hokkaido, Japan; mnako@den.hokudai.ac.jp

**Keywords:** bone, vasculature, endothelial cell, osteoblast, chondrocyte

## Abstract

Bone is a highly vascularized organ that not only plays multiple roles in supporting the body and organs but also endows the microstructure, enabling distinct cell lineages to reciprocally interact. Recent studies have uncovered relevant roles of the bone vasculature in bone patterning, morphogenesis, homeostasis, and pathological bone destruction, including osteoporosis and tumor metastasis. This review provides an overview of current topics in the interactive molecular events between endothelial cells and bone cells during bone ontogeny and discusses the future direction of this research area to find novel ways to treat bone diseases.

## 1. Introduction

Bones are highly vascularized, receiving more than 10% of the cardiac output [1,2]. Generally, bones consist of a firmly calcified exterior named cortical (or compact) bone and interiorly connected but structurally distinct trabecular (or cancellous) bone, which encases bone marrow (BM) with cells of hematopoietic and mesenchymal cell lineages. Therefore, bones not only support the body and organs but also provide stem cell niches for vascular and skeletal as well as hematopoietic cells, thus endowing the microstructure enabling these distinct cell lineages to reciprocally interact.

In flat bones, such as parietal and frontal bones, microvascular networks are well established in distinct tissue layers of periosteal and dural tissues, cortical bone, and BM to receive blood supply throughout the bone tissue. Long bones, exemplified by the femora and tibiae, receive blood supply from several major arteries, such as the central nutrient artery, the metaphyseal–epiphyseal arteries, and the periosteal arteries, which leads to the formation of vascular networks throughout the bone and BM and eventually drains blood via a large central vein. Therefore, the highly vascularized structure of bones enables bone-derived factors to circulate throughout the body, thus rendering bones endocrine organs [3].

In this review, we first discuss the current understanding of the relevance of vasculogenesis in skeletal patterning, bone morphogenesis, and regenerative bone morphogenesis after fracture repair. We further discuss the pathophysiological roles of vasculogenesis in bone-destructive diseases, such as age-related bone loss and tumor metastasis, to explore novel treatments against these diseases from the perspective of skeletal vascular interaction.

## 2. Vascularization during Initial Skeletal Patterning

Bone formation and vascularization concomitantly proceed in a manner of reciprocal regulation. However, these two distinct tissue developments appear to be independently patterned in the pharyngula, which consists of the embryonic head encasing the developing brain, trunk, and limbs. In this embryonic stage, mesenchymal tissues that eventually develop into skeletal elements and vasculature are essentially covered by single epithelial layers. Positioning of skeletal elements is initiated and induced by local signals from adjacent epithelial structures. Recent studies have proposed that the Turing’s reaction–diffusion mechanism of combinatorial molecular signals from adjacent tissue structures, such as epithelia, determine the exact number and spacing of bones and the positioning of joints in the limb and calvaria mesenchyme, which will be discussed in the next section [4,5].

Embryonic bone formation commences with the migration and localization of mesenchymal cells to future bone-forming sites followed by the condensation of mesenchymal cells [6]. The origins of mesenchymal condensates differ depending on embryonic positions: cranial neural crest cells in the skull and facial bones, including the maxilla and mandible; axial mesoderm cells in axial bones, such as vertebral bodies; and lateral mesoderm in limbs [4,7,8,9]. Each mesenchymal condensate then acts as a mold for future morphological and functional bone development.

Highly coordinated initial patterning of the skeleton and vasculature has long received close attention from developmental biologists and is well investigated using a central study model of limb development for tissue and organ patterning [10]. In the initial stages of limb development, in which embryonic epithelia merely cover the mesenchyme, vasculogenesis is concomitantly initiated by the formation of a vascular plexus embedded within the mesenchymal core. The origin of this vascular plexus is initially derived from sprouts of the dorsal aorta and then somite-derived angioblasts, while the skeletal mesenchyme originates from the lateral plate mesoderm. As skeletal mesenchyme condensation is initiated, the unpatterned vascular plexus regresses from the emerging cartilage anlagen and then develops into a highly branched and patterned vascular network. Importantly, avascularized areas of skeletal mesenchymal condensations emerge from previously vascularized regions as a result of vascular regression, creating a hypoxic environment in skeletal cells. Therefore, the initial positionings of avascular skeletal elements and their associated vascular mesenchyme territories are concomitantly patterned in a reciprocally complemental manner (Figure 1).

SOX9, an essential transcription factor of chondrogenesis, transiently induces vascular endothelial growth factor (VEGF) in the condensed mesenchyme, which regulates the morphogenesis of the vascular network in a long-range manner, as vascularization is not induced in the Vegf-expressing condensed mesenchyme. This observation demonstrates that early skeletal tissue regulates vascular patterning, thus providing a view of the skeletal mesenchyme as a signaling center for vascular patterning. However, possible reciprocal regulation by other molecules between the firming skeleton and its developing vasculature cannot be excluded since cross-regulation between developing tissues can imbue greater flexibility and robustness with the developmental program [11].

As an evolutionary question, it is unclear why the forming skeleton adopted an avascular environment by regulating vascular regression. Other organs, such as the lung, liver, kidney, and pancreas, develop concomitantly with their embedded vasculature. This appears to be reasonable, as the developing organs increase their demand for nutrients and oxygen. The regulated vascular regression and morphogenesis of the vascular network by the forming skeleton may provide a compensation mechanism for nutrient and oxygen supply, which facilitates synchronized interorgan development and undoubtedly provides a prime state for future bone development, as will be discussed in later sections. Avascular skeletal condensation appears to be an evolutionary constraint, which might have contributed to evolutionary divergence between organisms, as will be discussed in the next section.

## 3. Patterning and Positioning of Skeletal Elements

Spatiotemporal expression patterns of growth factors, such as BMPs, Wnts, Hedgehogs, FGFs, and families of transcription factors, such as Hox, Dlx, and Sox, provided fundamental views for understanding embryonic development and the patterning of tissues. The development of skeletal elements in the limbs and the craniofacial area has been described in the context of reciprocal epithelial–mesenchymal interactions through the regulation of these molecules. However, how the exact positioning of the skeletal element is determined within the mesenchyme is less clear.

Recent studies have proposed that Turing’s reaction–diffusion mechanism of these signaling molecules determines the number and position of bones in the limb and calvaria. The reaction–diffusion model is a mathematical model that can account for autonomic pattern formation during development [5,12]. Turing mathematically demonstrated that robust positional information could arise in a system by introducing multiple diffusible chemical factors controlling their synthesis and degradation of each other and proposed the idea that this system might provide a fundamental basis for the morphogenesis of living organisms.

The positional information model is another important principle for understanding the patterning of developing organisms (Wolpert 1969, 1989). In this model, diffusible molecules locally produced in a given position of developing tissue create a concentration gradient across a tissue, thus providing positional information that can allow for diverse cellular responses, depending on distinct concentrations of morphogens. This model gained experimental support and provided a fundamental basis for interpreting large amounts of spatiotemporal molecular expression during embryogenesis. Therefore, the positional information model has long aided in our understanding of developmental morphogenesis by providing static images of polarities and patterns across tissues. Applying multiple and interactive morphogens to reaction–diffusion models can form a variety of patterns, such as dynamic changes in waves, spots, stripes, and more complicated labyrinth patterns, in a self-organizing manner (Figure 1a).

Positioning of phalanges has been described as digit patterning in the development of limb buds. Recent combinatorial approaches of biological experiments and mathematical modeling have revealed self-organization mechanisms, rather than coordinated positional information, operated in the limb mesenchyme. One of the most currently successful models is a BMP-Sox9-Wnt Turing network (BSW) model proposed by Raspopovic et al. [13,14]. Sox9, an essential transcription factor for chondrogenesis, is specifically expressed in the condensing (avascular) mesenchyme, while the growth factors BMP and Wnt can be detected in the noncondensing (vascularized) mesenchyme of interdigit areas. Sox9 expression is experimentally proposed to be negatively and positively regulated by BMP and Wnt, respectively. The diffusion rate of BMP exceeds that of Wnt. These and other biological findings satisfy the requirement of positive and negative feedback loops in the Turing model. The BSW model initially provides heterogenic expression patterns of these three molecules and then induces autoregulation of stable and periodic expression patterns, thus establishing digit patterning. Additional molecules of FGF specifically expressed in the apical ectodermal ridge and Hox transcription factors expressed in the posterior mesenchyme can modify this BSW system [14]. The decreased function of FGF and Hox increases the number but decreases the size of forming digits [15]. Interestingly, further functional modulation of this system is proposed to be an important driving force for evolutionary divergence in digit patterning, as seen in fish fins, single digits in horses, double digits in camels, and quintuple digits in mice, humans, and other vertebrates [14,16].

A coupled Turing model of dot-forming and stripe-forming systems has been applied to the positioning of joints [17] (Figure 1b, right). This model can also explain the diversity of the number and position of joints in limbs and fins as well as misordered joint phenotypes observed in genetic mouse mutants. The Turing model integrated with computational biomechanics has also been applied to the patterning of cranial bones [18]. This model estimated the mechanical strain caused by the rapid expansion of the developing brain underlying the osteogenic mesenchymal layer of cranial neural crest cells. In that model, strain promotes pro-osteogenic molecules, such as BMP and Wnt, and, more interestingly, alters reaction/diffusion distances, thus resembling the actual pattern of cranial bone formation. Altering model parameters can also explain phenotypes of craniosynostosis of prematurely fused bones. These findings suggest that mathematical modeling with observable and measurable parameters will facilitate our understanding of more complex biological patterning events and interactions between sets of molecules.

## 4. Coupling of Osteogenesis and Vasculogenesis

Following this mesenchymal condensation, bone formation occurs by either of two processes: intramembranous ossification or endochondral ossification (Figure 2). In intramembranous ossification, cells in the mesenchymal condensate directly differentiate into osteoblasts to form flat bones, such as the skull and facial bones (Figure 2a). These flat bones consist of layers of compact bone, which are interspersed with BM. Differentiating mesenchymal cells secrete proangiogenic factors, such as VEGF-A, as well as osteogenic factors, which promote the differentiation of mesenchymal cells into osteoprogenitors and osteoblasts to form ossification centers, and the attraction of blood vessels to the ossification centers, which also promote osteogenesis [19].

VEGF signaling plays a critical role in intramembranous ossification [19,20,21,22]. Shortly before the initial ossification of flat bone, such as frontal and parietal bone, capillaries of a small diameter develop into the avascular mesenchymal layer that surrounds the mesenchymal condensation center, the future site of the initial ossification. The mesenchymal condensation center expresses VEGF, attracting endothelial cells (ECs), and the small vessels then invade the condensation center, where they almost concomitantly undergo mineralization. The continuous cascade of vascular invasion and mineralization expands externally in all directions. Therefore, bone patterning is associated with extensive development of the internal and external network of blood vessels.

In endochondral ossification, long bones develop through an intermediate stage of chondrocyte differentiation and avascular cartilage formation [23] (Figure 2b). Endochondral ossification generates the majority of bones, including long bones, such as the femur and tibia, and vertebral bodies, through which mesenchymal condensates differentiate into avascular cartilage that is eventually replaced by bone.

In both processes of ossification, vascular invasion and blood vessel growth into the mesenchymal condensate are important later events in bone development through the expression of VEGF, which enhances osteogenesis and supports vascular patterning during bone development [21,24,25]. As early as during mesenchymal condensation, vascular development in the peripheral mesenchyme proceeds by the expression of transforming growth factor beta 1 (TGFβ1) and its downstream effector connective tissue growth factor (CTGF), which can provide a poised state for vascular invasion into the bone-forming mesenchyme condensate [21].

The key event in vascular invasion into the developing bone tissue after mesenchymal condensation is hypoxia, which stabilizes hypoxia-inducible factor-1α (HIF-1α) subunits and activates downstream signaling pathways, including VEGF signaling [26,27]. In fact, HIF-1α loss-of-function mice exhibited decreased bone volume and bone vascularity [21]. VEGF signaling from the bone-forming mesenchyme plays a critical role in coupling vasculogenesis and osteogenesis [28]. During endochondral ossification, VEGF expressed in hypertrophic chondrocytes promotes vessel invasion and recruitment of chondroclasts, which enables the replacement of the cartilaginous template by naïve woven bone [20,29,30]. Vessel invasion accompanies osteoblast precursors expressing Osterix, an osteogenic transcriptional factor, which localize in a perivascular manner, indicating a strong relationship between vasculogenesis and osteogenesis [31].

In developing bone, osteoblasts participate in bone mineralization, laying down bone matrices in the vicinity of capillaries. In other words, the endothelium in the bone-forming site serves as a template for new bone formation, indicating that vascular patterning directs bone morphogenesis. The blood vessels distributed in the bone-forming sites are coated with collagen that is deposited by adjacent osteoblasts, thereby contributing to osteoid formation [32]. Lack of basement membrane with blood vessels in developing bone suggests the involvement of metalloproteinases (MMPs) in the degeneration of basement membrane. It is tempting to speculate that blood vessels may play more dynamic roles in bone patterning by regulating the formation and degradation of bone matrices.

Alterations in vascular growth cause compromised bone healing, osteonecrosis, osteoporosis, and nonunion fractures [33,34,35,36,37]. Therefore, vasculogenesis and osteogenesis are tightly coupled in the pathophysiology of bone, as will be discussed in later sections.

## 5. Changes in Endothelial Cells (ECs) Subtypes in Postnatal Bone Development

Distinct morphology and patterning of blood vessels depending on the bone site have long been recognized; however, until recently, the functional relevance and molecular characteristics of ECs in different bone regions were not well understood. Kusumbe et al. reported that capillary EC subtypes in BM might be able to be identified based on the marker expression and functional characteristics [38]. Two EC subtypes (H and L) were proposed to be distinguished by the expression of the cell-surface markers endomucin (Emcn) and CD31, the former of which is an EC population showing relatively high levels of CD31 and Emcn (CD31^hi^/Emcn^hi^ ECs) predominantly found in the metaphysis as well as in the endosteum (thin connective tissue layer covering the inner surface of compact bone), while the latter of which shows relatively low levels of these markers (CD31^lo^/Emcn^lo^ ECs) (Figure 3). These two subtypes comprise the sinusoidal vasculature of the diaphysis.

Langen et al. identified a third population of ECs (type E), which was observed to be abundant in embryonic and postnatal long bones [39]. Type E ECs were very similar to type H ECs (CD31^hi^/Emcn^hi^ ECs); however, they showed relatively low and high expression of Emcn and CD31, respectively, compared with type H ECs.

The proportion of these EC subpopulations within the bone dramatically changes during ontogeny. Type E ECs account for the majority in embryonic development, with some type L and only a small number of type H ECs present. In postnatal development, however, the proportion of type E ECs dramatically decreases, accounting for only 2.2% of bone ECs as of postnatal day 28 (P28). The proportion of type H ECs peaks at P6 and declines with time, while the proportion of type L ECs steadily increases throughout life [39]. A lineage analysis using a tamoxifen-induced fluorescent reporter of Apln-CreERT [40] demonstrated that type E ECs were able to differentiate into type H ECs and could further differentiate into type L and arterial ECs during postnatal development [39]. However, whether or not this lineage can contribute to venous vasculature in bone is not yet clear.

The distinct capillary EC subpopulations of types H and L localize to different regions of the bone and display a distinguished morphology from approximately P6 onward [39] (Figure 3). Type H vessels are primarily found in the metaphysis and periosteum, the latter of which shows a unique columnar structure. The type H columns are interconnected via loop-like arches at the distal edge, forming bud-shaped protrusions that extend toward the layer of hypertrophic chondrocytes in the cartilaginous growth plate [41]. Distal arteries and distal arterioles directly supply blood flow to the type H vessels of the metaphysis and endosteum. In contrast, type L ECs establish the sinusoidal vasculature of the diaphysis, which is not directly connected to the arteries and arterioles. These differences in spatial vasculature create regionally distinct perfusion, metabolic characteristics, and oxygen tension [41,42] and may provide site-specific differences in bone structure and metabolism.

These distinct capillary EC subpopulations can also be functionally distinguished by the presence of different perivascular cells. In the metaphyseal region, type H capillary-associated perivascular cells contain mesenchymal stem and progenitor cells expressing nestin and Sca1 [43]. Perivascular cells expressing PDGFRβ and NG2 are regulated by EC-derived PDGF-B [44] (Figure 3). In the diaphyseal region, sinusoidal type L vessels are associated with mesenchymal stem cells expressing leptin receptor (LEPR) and PDGFRα [45] as well as CXCL12- abundant reticular (CAR) cells [46] (Figure 3). Molecular signals from ECs and their associated mesenchymal cells maintain hematopoietic stem cell populations [45,46].

It can be relevant to know how highly angiogenic type H vessels transform into quiescent type L vasculature during the postnatal period and maturing bone development. Dzamukova et al. recently demonstrated that mechanical loading through increased body weight and muscle contraction triggered these skeletal vessel shifts [47]. Piezo1, a mechanoreceptor in osteoblasts, senses mechanical loading, which induces burst secretion of DMP1, an important extracellular matrix molecule regulating bone maturation. Large amounts of extracellular DMP1 inhibit VEGF signaling in the adjacent type H vessels, which lose their characteristic features and become type L vessels.

## 6. Pathophysiological Roles of Bone Vasculature in Age-Related Bone Loss: Osteoporosis

The vasculature of the skeletal system is essential for bone development, as discussed above, and plays important roles in bone homeostasis and pathology. The amount and shape of adult healthy bone are maintained through the balanced regulation of bone formation and resorption by osteoblasts and osteoclasts, respectively, which is recognized as bone remodeling. Bone remodeling involves the repair of microdamage to bone, the constant renewal of older bone, and calcium homeostasis. Dysregulation and dysfunction of bone homeostasis cause an imbalance between bone formation and resorption and can lead to a range of pathological conditions.

Bone density decreases with age, when the rate of bone resorption overtakes the formation rate [48,49,50,51]. Osteoporosis is a major age-related disease that occurs in both men and women and is characterized by a reduction in bone mass, disrupted microarchitectural integrity, and increased risk of fracture. Postmenopausal women show increased susceptibility to osteoporosis mainly due to the sudden decrease in estrogen levels. The decrease in sexual hormone levels (testosterone) in aged men is also associated with an increased risk of osteoporosis. Recent studies have demonstrated a strong interdependence of bone and vasculature in the pathophysiology of aged bone [48].

Age-related loss of type H ECs appears to be critical in the pathogenesis of osteoporosis [38,52,53] (Figure 4a). Postmenopausal osteoporosis is associated with a reduction in sinusoidal and arterial capillaries, thus leading to reduced bone perfusion and oxygen levels. An associated reduction in bone vasculature and bone-forming cells was seen in ovariectomized (OVX) mouse models of osteoporosis [54,55]. Furthermore, a significant reduction in type H vessels is also observed in OVX mice [53,56]. The clinical relevance of this finding is supported by the decline in human type H endothelium observed in postmenopausal women [57].

In a mouse model of glucocorticoid-induced osteoporosis (GIO), reduced type H vessels, mature osteoblasts, and osteocytes were also observed [55], which appears to be compatible with the observations that glucocorticoids decrease blood flow and inhibit angiogenesis by reducing VEGF levels [55,58,59,60]. Taken together, these findings suggest the importance of maintaining type H vasculature to treat osteoporosis in human patients.

## 7. Roles of Bone Vasculature in Regenerating Bone after Fracture Repair

Bone fracture is one of the most common traumatic injuries in humans and damages not only the bone architecture but also surrounding soft tissues, including vasculature. Even in adult humans, bone can be fully regenerated, with vasculature playing a critical role in this process [61,62,63] (Figure 4b). Bone fracture healing involves multistep processes, and damaged blood vessels cause hemorrhaging that attracts cytokine-secreting inflammatory cells to form a fibrinous clot [64]. This proinflammatory state stimulates cell proliferation and differentiation via the expression of IL-1 [65], MMP-9 [66], and BMPs [67,68] to form a soft callus that stabilizes the injury site [63]. Hypoxic conditions and high lactate levels in the fracture site functionally activate HIF-1α and its downstream target VEGF, which stimulate angiogenesis and osteogenesis, as described above, through which vascularized hard callus replaces the formed soft callus [21,62,63,68]. Indeed, functional loss of HIF-1α in the osteoblastic lineage delays callus formation and impairs fracture healing [21].

Angiogenesis is considered to be essential in fracture repair [69]. During the repair phase, VEGF stimulates the regrowth of blood vessels into the site of injury to restore normal oxygen and nutrient supply and activate the osteoblast function [62]. While inhibition of VEGFR1 and VEGFR2 impairs osteogenesis and chondrogenesis and reduces callus formation [70], VEGF administration significantly accelerates fracture repair [71]. TNF-α administration has also been shown to promote fracture repair by recruiting muscle-derived stromal cells and promoting osteogenic differentiation [72]. Regrowth of sensory nerve fibers is stimulated by NGF, creating pain sensation. NGF also stimulates VEGFA-mediated revascularization and promotes ossification via TrkA-mediated communication between sensory nerves and osteoblasts [73,74]. Inhibition of TrkA signaling reduces nerve regrowth and revascularization, delaying the ossification of fracture calluses [74]. It has been noted that blood supply is essential for callus formation and fracture repair [75,76]. Clinically, the condition in 10% of patients with compromised fracture healing is reportedly due to a lack of blood supply after injury [77,78]. Pharmacological inhibition of angiogenesis prevents callus formation and fracture healing and may cause abnormal healing by the formation of fibrous tissue [69]. Proper angiogenesis during bone repair is not only required to supply oxygen and nutrients to the healing sites but is also essential to enable osteochondro precursors to migrate into the fracture callus [31].

Similar to skeletal development, revascularization of the fracture site involves an ossification process through either intramembranous or endochondral ossification; in the former, new bone tissue is formed directly via progenitor differentiation into osteoblasts, while in the latter, new bone is indirectly formed via cartilage intermediates [79]. Generally, stable fracture with a sufficient supply of oxygen, nutrients, and trophic factors allows intramembranous ossification, whereas unstable hypoxic fractures are repaired through endochondral ossification [80].

Through the remodeling phase, the fracture site reduces newly formed immature woven bone, hard callus, and vessels to restore structurally and mechanically mature bone as preinjury cortical or trabecular bone [63]. A complex but well-organized interaction between osteoclasts, osteoblasts, and vasculature drives the remodeling phase, in which proinflammatory cytokines, such as IL-1 and TNF-α, play important roles [63,81]. Blockade of angiogenesis significantly perturbs this phase [82]. Slit homologue 3 proteins (SLIT3) is an axon guidance molecule that induces the migration of endothelial cells. Slit3-deficient mice showed reduced type H ECs and impaired fracture repair, suggesting a role of type H ECs in fracture repair [83]. Taken together, these findings indicate that proper angiogenesis and angiogenic regulation throughout bone repair are mandatory for repatterning and recovering mature functional bone. Therefore, angiogenic regulation of skeletal lineage cells can be targeted for regenerative medicine of bone repair.

## 8. Pathological Roles of Bone Vasculature in Cancer Metastasis

Bone is one of the major target organs for cancer metastasis along with the lung and liver. Bone metastasis can induce chronic bone pain and diminish bone rigidity, thus increasing the risk of bone fractures, which eventually reduce the quality of life of cancer patients. Bone metastasis can also develop hypercalcemia, which causes constipation, nausea, thirst, and loss of appetite. The bone microenvironment facilitates the survival and activation of cancer cells due to its enriched vasculature and nutrient factors stored in the bone matrices. Cancer cells can even alter their phenotype to adapt to the bone environment and further modify the bone microenvironment for their survival and growth.

The first step of bone metastasis is the extravasation of circulating cancer cells and homing to the BM. Many factors are thought to play a role in attracting tumor cells to the bone microenvironment, including the CXCL12/CXCR4 signaling axis. For example, CXCL12 is expressed in many types of cells in bone environments, including mesenchymal stem cells and endothelial cells, and attracts CXCR4-expressing cancer cells [84]. Following disseminated tumor cell (DTC) homing to bone, only a limited number of DTCs can survive [85]. Recent studies have revealed the mechanism of control of cancer cell dormancy through angiocrine signals [86,87].

The highly vascularized structure of the BM can provide a niche for cancer cells as well as hematopoietic and mesenchymal stem cells by facilitating crosstalk between cancer cells and ECs [44,86]. VEGFR1-positive BM hematopoietic progenitors can attract cancer cells, thus initiating a premetastatic niche. Thrombospondin-1 produced by ECs helps establish a stable BM vascular niche for DTCs. After being integrated into this niche, DTCs can maintain their dormant state over a long period [88]. This DTC dormancy is facilitated by a microenvironment of low sinusoidal blood flow around the large vessel, which also provides therapy resistance conditions [87].

Interaction between DTCs and ECs plays a role in DTC dormancy and therapy resistance. Coculture of acute myeloid leukemia (AML) cells with BM ECs increases the proportion of quiescent AML cells [89]. Molecular interaction between EC-derived von Willebrand factor and VCAM1 and integrin expressed by DTCs is crucial for DTC chemoresistance, demonstrated by observations that integrin-blocking antibodies sensitize DTCs to chemotherapy and prevent bone metastasis [90]. EC-derived PDGF-B signaling functions as another key regulator of tumor cell dormancy and therapy resistance by acting on pericytes that secrete quiescence-inducing factors, such as CXCL12 [87]. DTC dormancy appears to be achieved by mimicking microenvironmental cues for normal HSC dormancy [86,91].

Dormant cancer cells become reactivated after an extended period of dormancy. DTC reactivation is triggered by microenvironmental changes. These cells can modulate angiogenesis by producing proangiogenic factors, such as VEGFA to induce angiogenesis [92]. Such switching of angiogenesis induced by cancer cells is necessary for allowing cancerous lesions to expand beyond a few millimeters in size [93]. In the BM environment, oxygenation is quite low despite high vascularization. The adaptive response to changes in tissue oxygenation is mediated through HIFs. Stabilized HIF-1α causes VEGF upregulation, leading to angiogenesis. Therefore, hypoxia, a characteristic of the bone microenvironment, may have a role in DTC reactivation. VEGF promotes MMP production [94], and MMP9 induces angiogenesis coupled with bone resorption, which may affect dormant cell reactivation [95].

In addition, the aging of BM vasculature plays a role in DTC proliferation. The age-related reduction in perivascular PDGF-B signaling, which is a key event for age-related loss of type H vessels as discussed above, reactivates the proliferation of dormant DTCs [87,96] (Figure 4c), which may result in a microenvironment facilitating metastasis, even after decades of latency [44,87]. The interactions between cancer cells and the bone microenvironment are crucial in tumor cell dissemination. The vasculature is a key component of the bone microenvironment for cancer cells and contributes to bone metastasis. Understanding the importance of vasculature for bone metastasis is essential for developing novel treatment modalities to prevent or cure bone metastasis.

## 9. Future Perspectives

As we discussed in this review paper, the spatiotemporal interaction between endothelial cells and neighboring skeletal cells plays critical roles in development, homeostasis, and pathological bone destruction. Spatial transcriptomics technologies developed in recent years may be promising tools for further clarifying skeletal–vascular interactions in many biomedical aspects, leading to significant breakthroughs in tissue development, tissue architecture, and disease research [97,98]. Spatial transcriptomics technologies allow transcriptomic information to be obtained from intact tissue sections, with which various biological insights, such as interaction between cells even at single-cell resolution, larger-scale tissue interaction, and organ-level topological regulation, can be achieved in a more holistic and consecutive manner. Baccin et al. identified sources of pro-hematopoietic factors on the surfaces of sinusoidal and arteriolar vessels, namely perivascular microniches for HSC maintenance and differentiation in the context of three-dimensional BM organization [98]. Advances in spatial omics such as spatial transcriptomics, epigenomics, proteomics, and metabolomics, and their integrations at various biological scales, will enable us to decipher previously uncovered pathophysiological mechanisms in bone, and help refine classifications of certain diseases, facilitating precise and individualized medical treatment for patients. These advances will also require further innovations in data analyses as data scale and complexity increase. Artificial-intelligence-driven analyses can be integrated with mathematical modeling, such as Turing’s model, which will aid in the interpretation of more convoluted molecular interactions unveiled by spatial omics (Figure 5).

## 10. Conclusions

As discussed in this review paper, the function and dysfunction of bone vasculature underlie a wide range of aspects in the pathophysiology of the skeletal system during ontogeny that involve skeletal development, homeostasis, age-related comorbidities, and destruction by cancer. Emerging evidence suggests the importance of unveiling further skeletal–vascular interactions to treat human patients suffering from cancer as well as other bone-destructive diseases. The next breakthrough in skeletal medicine would come from a view of a self-organization system of a set of molecules across distinct biological scales, not from merely dissecting changes in the expression of the molecules involved. Advances in spatial omics and data analyses will help identify more complicated pathophysiological processes (Figure 5).

## Figures and Tables

**Figure 1 ijms-24-10912-f001:**
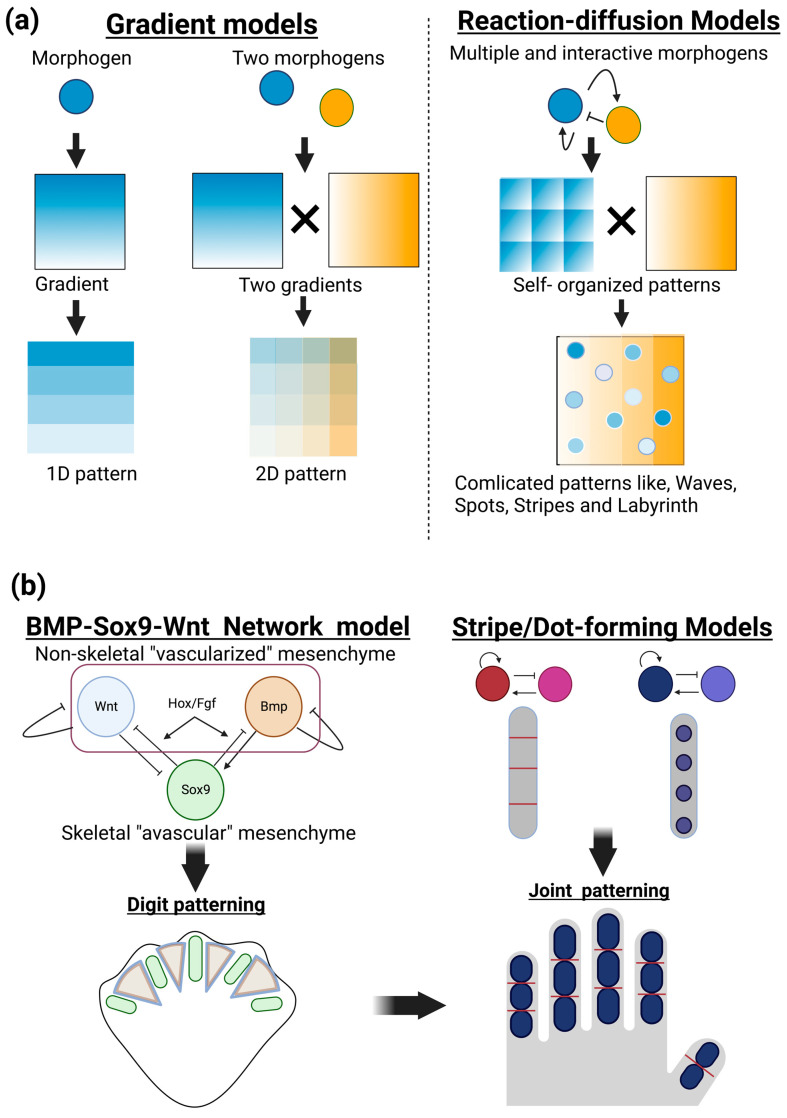
Patterning and positioning of skeletal elements by reaction–diffusion mechanisms. (**a**) Gradient models and reaction and diffusion models. In the gradient models (left), a given diffusible molecule (morphogen) locally produced in developing tissue forms a concentration gradient across tissue (1D pattern). Additional morphogens can form squared patterns containing distinct levels of morphogens in each area that endow different properties of future biological events (2D pattern). In the reaction–diffusion models (right), multiple and interactive morphogens can create a variety of biologically observable patterns, such as dynamic changes in waves, spots, stripes, and more complicated labyrinth patterns, in a self-organizing manner. (**b**) Patterning and positioning of skeletal elements by reaction–diffusion mechanisms. The BMP-Sox9-Wnt Turing network (BSW) model can explain digit patterning (left). Sox9, an essential transcription factor for chondrogenesis, is specifically expressed in the condensing (avascular) mesenchyme, while the growth factors BMP and Wnt can be detected in the noncondensing (vascularized) mesenchyme of interdigit areas. Sox9 expression is experimentally proposed to be negatively and positively regulated by BMP and Wnt, respectively. The BSW model initially provides heterogenic expression patterns of these three molecules, and their inter- and autoregulations create stable and periodic expression patterns, thus establishing digit patterning. A coupled dot- and stripe-forming reaction–diffusion system has been applied to the positioning of the joints (right).

**Figure 2 ijms-24-10912-f002:**
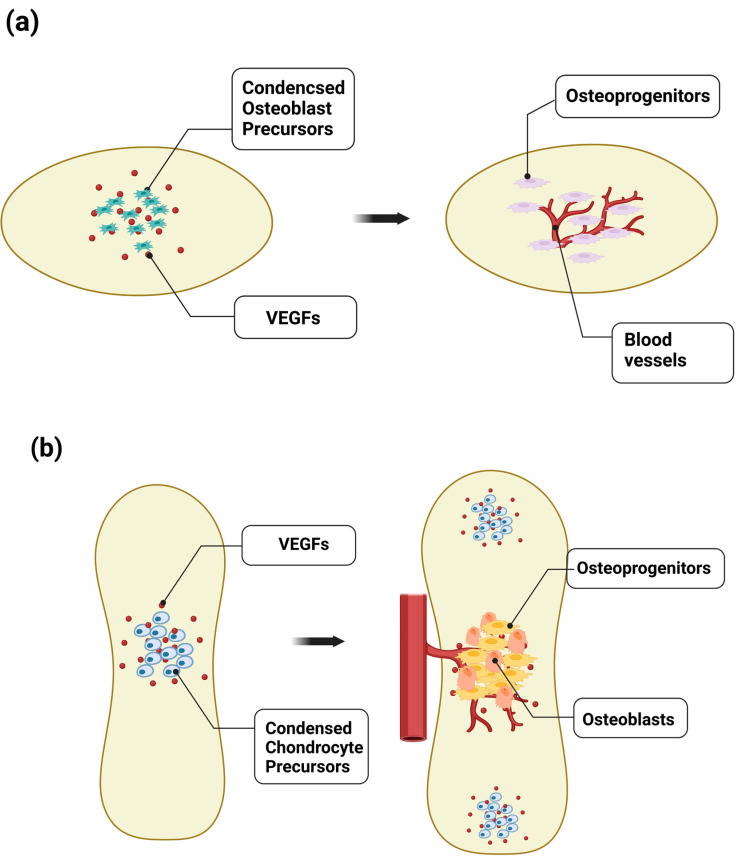
Disruption of avascular skeletal anlagen for initial bone development. (**a**) During endochondral ossification, VEGF expressed in hypertrophic chondrocytes promotes vessel invasion and recruitment of chondroclasts, which enables the replacement of the cartilaginous template by naïve woven bone. Vessel invasion accompanies Osterix-expressing osteoblast precursors that localize in a perivascular manner, indicating a strong relationship between angiogenesis and osteogenesis. (**b**) VEGF signaling also plays a critical role in intramembranous ossification. Shortly before the initial ossification of flat bone, such as frontal, parietal, and jaw bones, the mesenchymal condensation center expresses VEGF, attracting endothelial cells. The small vessels then invade the condensation center, which almost concomitantly undergoes mineralization, indicating the essential role of the vasculature in bone formation by promoting the differentiation of osteoblasts.

**Figure 3 ijms-24-10912-f003:**
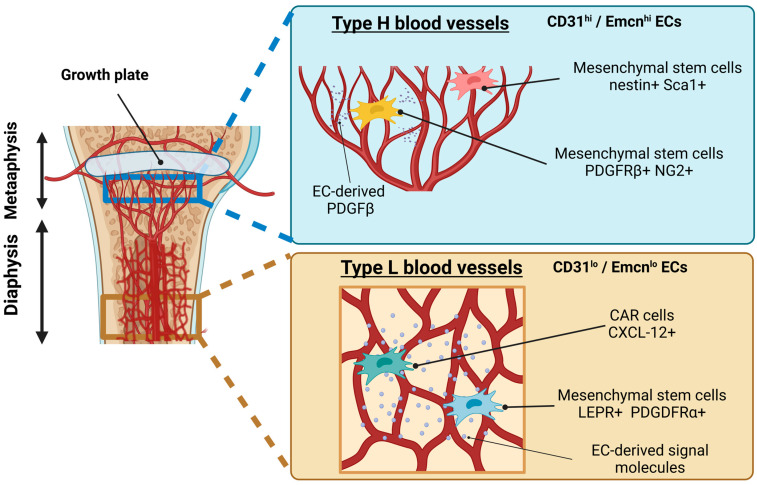
Type H and L vasculatures in developing bone. Two endothelial cell (EC) subtypes, type H and type L, are distinguished by the expression of the cell-surface markers CD31 and endomucin (Emcn). Type H (CD31^hi^/Emcn^hi^) ECs are predominantly found in the metaphysis as well as in the endosteum. Type L (CD31^lo^/Emcn^lo^ ECs) ECs compose the sinusoidal vasculature of the diaphysis. Type H vessels harbor mesenchymal stem cells, as indicated by providing pro-MSC factors, such as PDGF. Type L vessels fill niches for supporting MSCs as well as hematopoietic stem cells (HMCs) by secreting signal molecules. In postnatal development, type H ECs gradually lose their properties and population and convert into type L ECs.

**Figure 4 ijms-24-10912-f004:**
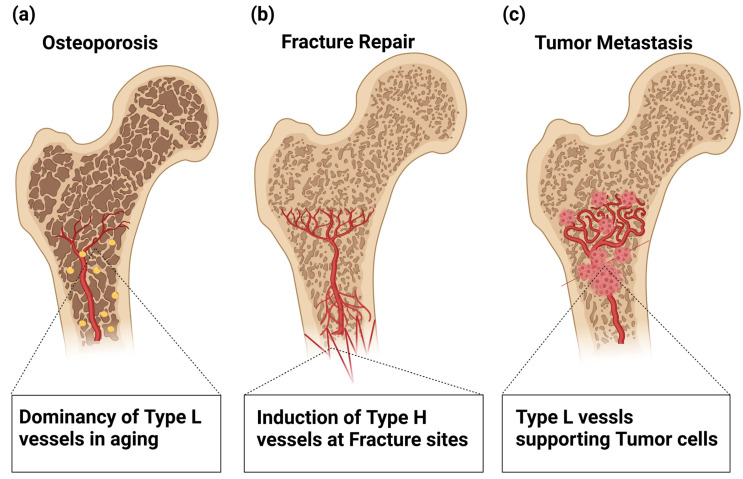
Involvement of Type H and L vasculatures in bone pathophysiology. (**a**) Age-related loss of type H ECs is involved in the pathogenesis of osteoporosis. (**b**) In fracture repair, hypoxic conditions in the damaged tissues trigger VEGF production that stimulates the regrowth of Type H vessels into the fracture sites, which restores the normal oxygen and nutrient supply and activates osteoblast differentiation. (**c**) Age-related loss of type H ECs plays a role in dormant tumor cell (DTC) proliferation. The age-related reduction in perivascular PDGF-B signaling reactivates the proliferation of DTCs, thus providing a microenvironment facilitating metastasis even after decades of latency.

**Figure 5 ijms-24-10912-f005:**
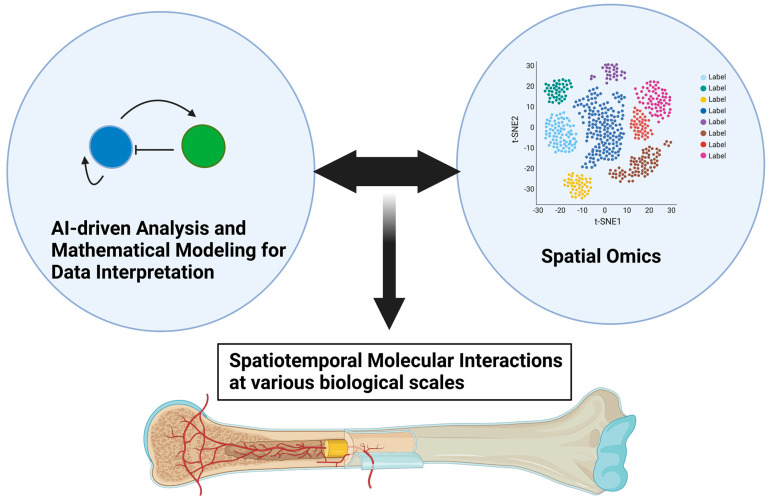
Advances in spatial omics and mathematical modeling as data interpretation for further unveiling skeletal–vascular interactions. Advances in spatial omics technologies will enable us to dissect various biological events into molecular interactions at resolutions of single cells, larger-scale tissue, and the organ level in a more spatiotemporally comprehensive manner. Mathematical modeling, such as that using Turing’s model, may aid in interpreting increasingly large data sets of molecular information at various biological scales.

## Data Availability

Not applicable.

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
