# Peer review of "Skeletal-Vascular Interactions in Bone Development, Homeostasis, and Pathological Destruction"

_ijms, 2023, doi:10.3390/ijms241310912_

Round 1
Reviewer 1 Report
The submitted manuscript, on “Skeletal-vascular interactions…” is very well prepared and informative. The evolutionary context and the historic contributions of Turing and Wolpert are welcome aspects. The core message is clear and novel, to my knowledge. It was a pleasure to give it a close reading. My suggestions for improvements are relatively minor. Perhaps of most concern are those relating to figure 3.
The English language usage is excellent. Places where change is needed are few:
· Line 2 of the text, the meaning of “hardly” is unclear. Does “hardly calcified” mean that it is barely calcified, or that it is firmly calcified?
· Page 1, line 34, the authors should be consistent in their pluralizations: either femora and tibiae, or femurs and tibias. I would prefer the former.
· 7th page, line 231, I do not understand “developed into the lost avascular…” Do the authors mean ‘last’ or something else?
The acronym VEGF first appears on page 3, line 98. It is not written out in full, although on page 6 it appears after “vascular endothelial growth factor.” This term should appear when the short form is first used.
The term “osterix” appears twice (pages 5 and 6). A definition would be welcome.
Page 7, beginning a new section, please write out endothelial cells (EC) in the heading.
Figure 3: The discussion of developing bone is crucial to the story line. The outline of a long bone in the figure is of a mature bone. The bone needs to be re-drawn so that there is a clear metaphyseal region between diaphysis and epiphysis.
Page 10, line 341, I question the validity of the claim that “Recent” studies have demonstrated …Most dates of the citations in this section are not that recent. Indeed, I was a little disappointed that the epidemiological/clinical publications cited here are so old. For example, surely there are data newer than 2012 that document bone density decrease with age.
Page 11, lines 421-424: These two sentences could use some citations, and again there is that word “recent.” The sentence “Recent studies have revealed…” should be supported by citations from within the past 5 years.
Page 12, figure 5 and page 13, lines 484 and subsequent: Use of the term “omics” without elaboration is daunting to the reader. In the text, the authors make clear that they are referring to many perspectives within the branches of science known informally as omics. The label to the figure refers specifically to spatial omics. Do they see “spatial omics” as an umbrella term? Readers would be helped if the authors were less jargon-heavy here. Somewhere (text or figure legend) they should say that they refer to study of the sum, or entire complement, of constituents within skeletal-vascular interactions (“omics”).
Pages 13 and 14: The publisher’s expectations have not yet been addressed in sections including Author Contributions, Institutional Review board Statement, Informed consent statement, and Data availability statement. Since this is a review paper, some of these sections may be waived, but at present they remain unaddressed.
see comments above. The language usage is very good.
Author Response
Letter to Reviewer 1
We would like to express our sincere thanks to the Reviewer 1. We have addressed all of your comments. Our changes were all highlighted in the revised manuscript.
Line 2 of the text, the meaning of “hardly” is unclear. Does “hardly calcified” mean that it is barely calcified, or that it is firmly calcified?
(Answer) We changed this word to “firmly” as suggested.
Page 1, line 34, the authors should be consistent in their pluralizations: either femora and tibiae, or femurs and tibias. I would prefer the former
(Answer) We changed this words to “femora and tibiae” as suggested.
7th page, line 231, I do not understand “developed into the lost avascular…” Do the authors mean ‘last’ or something else?
(Answer) We removed s “lost” from this sentence. It was really confusing, we admitted.
The acronym VEGF first appears on page 3, line 98. It is not written out in full, although on page 6 it appears after “vascular endothelial growth factor.” This term should appear when the short form is first used.
(Answer) This point was corrected as “vascular endothelial growth factor (VEGF)” in the first appearance.
The term “osterix” appears twice (pages 5 and 6). A definition would be welcome.
(Answer) This point was corrected as “osteoblast precursors expressing Osterix, an osteogenic transcriptional factor” in the first appearance.
Page 7, beginning a new section, please write out endothelial cells (EC) in the heading.
(Answer) This point was corrected as suggested.
Figure 3: The discussion of developing bone is crucial to the story line. The outline of a long bone in the figure is of a mature bone. The bone needs to be re-drawn so that there is a clear metaphyseal region between diaphysis and epiphysis.
(Answer) We totally agree with this. We revised the figure with re-drawing the bone having a clear cartilaginous growth plate that indicates a metaphyseal region.
Page 10, line 341, I question the validity of the claim that “Recent” studies have demonstrated …Most dates of the citations in this section are not that recent. Indeed, I was a little disappointed that the epidemiological/clinical publications cited here are so old. For example, surely there are data newer than 2012 that document bone density decrease with age.
(Answer) We replaced a reference in this sentence as [48], accordingly to [48-57] in the related sections.
Page 11, lines 421-424: These two sentences could use some citations, and again there is that word “recent.” The sentence “Recent studies have revealed…” should be supported by citations from within the past 5 years.
(Answer) We replaced a reference in this sentence as [86, 87], accordingly.
Page 12, figure 5 and page 13, lines 484 and subsequent: Use of the term “omics” without elaboration is daunting to the reader. In the text, the authors make clear that they are referring to many perspectives within the branches of science known informally as omics. The label to the figure refers specifically to spatial omics. Do they see “spatial omics” as an umbrella term? Readers would be helped if the authors were less jargon-heavy here. Somewhere (text or figure legend) they should say that they refer to study of the sum, or entire complement, of constituents within skeletal-vascular interactions (“omics”).
(Answer) We rephrased this section more specifically for better understanding to the readers. Rephrased sentences are now highlighted in the revised text.
Pages 13 and 14: The publisher’s expectations have not yet been addressed in sections including Author Contributions, Institutional Review board Statement, Informed consent statement, and Data availability statement. Since this is a review paper, some of these sections may be waived, but at present they remain unaddressed.
(Answer) Thanks for this comment. We added description for these points in the revised text.

Reviewer 2 Report
This is a well summarized and well written review manuscript covering the topics of bone-vascular interactions in development, homeostasis, and pathological conditions. There are a few minor comments that need to be addressed prior to publication.
1. The sentences in lines 71-75 should be referenced.
2. The FGfs in line 120 should be FGFs.
3. The order of endochondral and intramembranous ossification in line 389 should be reversed. Intramembranous should come first, followed by endochondral based on the following sentences.
Author Response
Letter to Reviewer 2
We would like to express our sincere thanks to the Reviewer 1. We have addressed all of your comments. Our changes were all highlighted in the revised manuscript.
- The sentences in lines 71-75 should be referenced.
(Answer) We added these references [4, 5] in this sentence.
- The FGfs in line 120 should be FGFs.
(Answer) We corrected to “FGFs”.
- The order of endochondral and intramembranous ossification in line 389 should be reversed. Intramembranous should come first, followed by endochondral based on the following sentences.
(Answer) We corrected to this point, accordingly.
